# Experimental and theoretical investigation on the conductivity of complex fracture network in unconventional gas reservoirs

**Jinjian Gao** [1]*, **Lanxiao Hu**[2]

**1** Airport Department, Jiangxi Flight University, Nanchang, Jiangxi, China, **2** College of Energy Resources, Chengdu University of Technology, Chengdu, Sichuan, China

* gaojinjian1990@163.com

## Abstract

With the depletion of conventional oil and gas resources, unconventional oil and gas resources have become crucial as alternative hydrocarbon sources. Unconventional gas reservoirs typically have low porosity and permeability, requiring reservoir stimulation to create complex fracture network. Unlike single fracture, the fracture network exhibits complicated structures due to multiple fractures and their various distribution manners. To study the conductivity of complex fracture network, a new conductivity chamber is designed based on the discretized fracture network model. Experiments and orthogonal analysis are conducted to investigate the effect of critical influential factors such as fracture network structure, fracture height, proppant concentration, proppant diameter, and proppant type on the conductivity of complex fracture network. The experimental results demonstrate that compared to the single fracture, the conductivity of T-shape, F-shape, E-shape, ╪-shape, and ╁-shaped fracture networks increased by 49%, 131%, 150%, 188%, and 201%, respectively. Closure pressure and secondary fracture distribution significantly impact the conductivity of complex fracture network. Proppant diameter and proppant concentration also have substantial effects. Compared with existing experimental studies focusing on single fracture conductivity, this study presents experimental research on the conductivity of complex fracture network, thereby addressing a research gap in this field. Through synergistic integration of the Kozeny-Carman hydrodynamic theory with the hydraulic-electrical analogy principle, an innovative model of fracture network conductivity is formulated. The advantage of the proposed fracture network conductivity model lies in its systematic integration of theoretical analysis with experimental validation methodologies, effectively bridging the gap between computational predictions and laboratory observations. This study provides valuable insights for optimizing volume fracturing in tight reservoirs, facilitating their rapid and efficient development.

**Data availability statement:** All relevant data are within the manuscript and its Supporting information files.

**Funding:** This work was supported by the Natural Science Foundation of Sichuan Province (2023NSFSC0945 and 2025ZNSFSC1188) and Core Technology Project of China National Petroleum Corporation (2023ZG18) to H.L.

**Competing interests:** The authors have declared that no competing interests exist.

# 1 Introduction

Unconventional gas reservoirs generally have no natural productivity without fracturing. Hydraulic fracturing has become one of the key technologies in the development of unconventional gas reservoirs, enabling them to achieve industrial gas flows and making the development of unconventional gas reservoirs feasible [1–5]. Typically, the fractures formed in conventional reservoirs are single, symmetrical, bi-wing fractures. However, unconventional gas reservoirs have many natural fractures and possess high brittleness [6–8], which tend to create complex fracture network rather than the single fracture. These observations have been extensively found in the North American hydraulic fracturing practices [9–13]. During the fracturing process, brittle rocks and natural fractures continuously open or shear, forming a complex network of artificial and natural fractures. This approach leads to a substantial enhancement of the stimulated reservoir volume [14–17]. The technique to enhance production and ultimate recovery by creating such complex fracture network is known as volume fracturing, which plays an increasingly important role in the development of unconventional oil and gas reservoirs.

It is widely acknowledged that fracture conductivity is a key parameter for volume fracturing. In recent years, experiments of fracture conductivity have evolved from single proppant type and size to multiple types and mixed sizes. Wen et al. [18] pioneered the application of the long-term fracture conductivity instrument in China to study proppant embedding by using formation cores. However, the research object was a single fracture. Lu et al. [19] simulated fracture networks using various grooves with rectangular cross-sections. They examined the conductivity of five different network structures: "−", "T", "L", "F", and "╪". It proved that the fracture network created by volume fracturing had different influence on fracture conductivity. This experiment ignored the crushing and embedment of proppants. Wen et al. [20] designed a complex fracture network simulation device to study proppant settlement in "−", "T", "H", and "TF" network configurations based on the discretized fracture network model. This provides a foundation for the experiment in this article. Liang et al. [21] evaluated the fracture conductivity with various ratios of sands to ceramic proppants by the use of standard Fracture Conductivity Evaluation System under different effective closure stresses. Wang et al. [22] conducted triaxial experiments to explore the effect of displacement, stress difference, and weak surfaces on fracture conductivity. Shi et al. [23] examined the fracture conductivity of tight dolomite reservoirs, analyzing various influencing factors. They found that proppant diameter and proppant concentration significantly impacted fracture conductivity. Katende et al. [24] employed experimental and analytical simulation methods to study proppant embedment and fracture conductivity within Caney shale fractures. Improved fracture conductivity was attributed to proppant diameter. Liu et al. [25] conducted experiments on long-term non-uniform sand spreading in hydraulic fractures of tight reservoirs, considering different influencing factors. Based on the conductivity loss mechanism, the calculation model of the long-term conductivity of heterogeneous artificial fractures was established.

To guide the hydraulic fracturing field, the single hydraulic fracture conductivity model has been established. There are two types of models, one is an empirical

model based on experimental results, and the other is a theoretical model derived from fluid mechanics in porous medium and capillary model. Li et al. [26] created a new mathematical model to calculate fracture conductivity and proppant embedment. Moreover, factors affecting the fracture conductivity have been analyzed including proppant diameter, closure pressure, and proppant type. Liu et al. [27] proposed an innovative fracture conductivity calculation model based on a "fracturing efficiency index". Liu et al. [28] established a new model for predicting fracture conductivity considering the particle size redistribution caused by proppant crushing based on the Kozeny–Carman equation. However, the most important point has been overlooked: the fracture network is composed of multiple fractures.

In previous research, experimental research on fracture conductivity has mainly focused on single fracture. The experimental instrument for fracture conductivity is still a traditional diversion chamber which cannot meet the requirement of testing the conductivity of complex fracture network. There is very little research on the calculation model of the conductivity of complex fracture network.

Firstly, a new diversion chamber is designed and manufactured based on the discrete fracture network model. Then, rock slabs are combined into fracture network. Experiments with different fracture structures are conducted using a new type of diversion chamber. Orthogonal experiments are used to study the primary and secondary factors affecting the conductivity of complex fracture network. Finally, based on the Kozeny-Carman equation and the principle of hydraulic-electrical analogy, a new model of fracture network conductivity is established.

## 2 Experimental methods

Traditional conductivity measurement chambers can only test the conductivity of a single fracture, falling short of testing the conductivity of complex fracture network. In this study, we propose a new conductivity measurement chamber based on the experimental observations obtained from the settlement and transportation of proppant in the complex fracture network and the discrete fracture network model [29,30]. This new chamber features a larger inner cavity space enabling the investigation of the fracture network and can be integrated with the fracture conductivity measurement setup.

Fig 1 illustrates the internal structure of the new conductivity measurement chamber with a dimension of 170 mm × 90 mm × 50 mm. The chamber is equipped with stainless steel pads to adjust the fracture height. The compressive strength of stainless steel is 520 MPa. Various shapes of rock slabs simulate different fracture networks. The surrounding space is sealed with foam rubber. Four interfaces are implemented for monitoring the inlet and outlet pressures, alongside flow rate measurements. Compared with the standard conductivity measurement chamber, the new proposed chamber possesses the following features:

(1) The diversion chamber, constructed from stainless steel, undergoes wire cutting internally to ensure smooth cavity walls.

(2) The rock slab is positioned 10 mm from the edge wall on one side. This arrangement ensures even gas distribution at the fracture's inlet and outlet, mitigating the end effect and enhancing test accuracy.

(3) Due to the application of sealant on both sides of the rock slab, pressure measurement on the walls is not feasible. Pressure measuring ports are located on the inlet and outlet side walls to obtain the pressure differences. The effective fracture length is 160 mm.

(4) Filter plugs are installed in the gas inlet and pressure measuring ports to prevent the leakage of the crushed proppant during the experimental process.

(5) The maximum closure pressure is 100 MPa.

The experimental instrument primarily comprises six components, as illustrated in Fig 2.

A

B

C

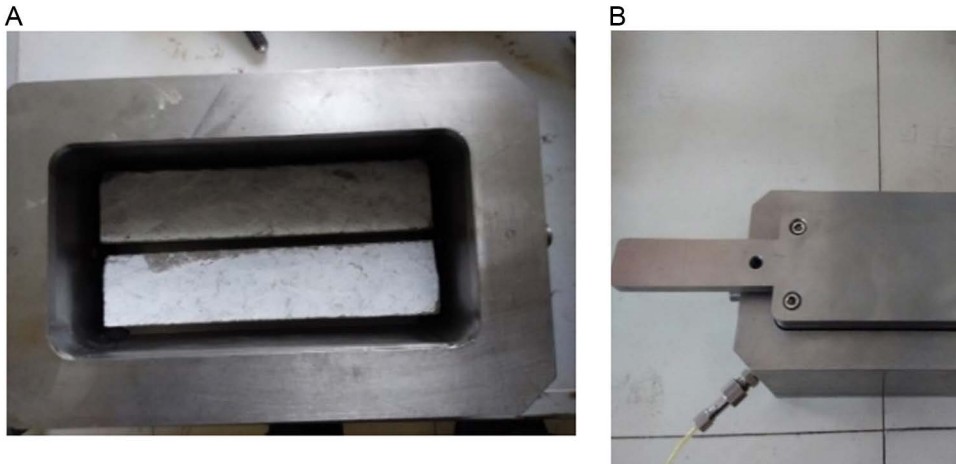

- Upper sealing plate
- Fracture height
- Rock slab
- Main body
- Lower sealing plate

**Fig 1. Structure diagram of new conductivity measurement chamber.** (a) Internal diagram, (b) External image, (c) Explosion view.

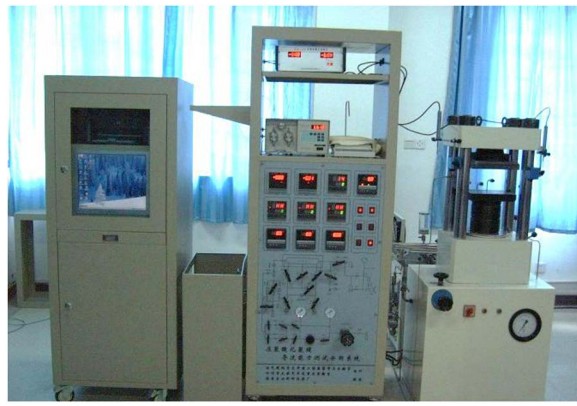

**Fig 2. Fracture conductivity instrument of FCES-100.**

This investigation employs the volumetric equivalence theorem, by which the void volume within the fracture network is equated to that of an imaginary single fracture's volume. The fracture network width is assumed to be the same as the width of the imaginary single fracture. This width is calculated by dividing the total volume by the surface area parallel to the seepage direction:

$$w_f = V/S \tag{1}$$

Where $w_f$ is the equivalent fracture width (cm), $V$ is the proppant volume (cm³), $S$ is the horizontal area of the fracture network (cm²).

The calculation equation for fracture conductivity can be written as:

$$kw_f = \frac{Q\mu L w_f}{A_f \Delta P} \tag{2}$$

Where $k$ is permeability (µm²), $Q$ is the flow rate (cm³/s), $\mu$ is gas viscosity (mPa·s), $L$ is the fracture conductivity length (cm), $A_f$ is the cross-section area of fracture network (cm²), $\Delta P$ is the pressure difference between inlet and outlet (0.1MPa).

We construct several typical fracture network structures based on the discretized fracture network model. Fig 3 summarizes these structures, including "−" type, "T" type, "F" type, "E" type, "+" type, "╪" type, and "Ⱶ" type. For the conductivity measurements, the width of secondary fractures is assumed to be equivalent to that of the main fracture.

Taking the "−" type fracture network as an example, the experimental scheme for changing the proppant concentration is as follows: the fracture height is 50 mm; and the proppant concentrations are 1.5, 2, 2.5, 3, 3.5, and 4 kg/m². In hydraulic fracturing of unconventional reservoirs, optimized stimulation protocols typically utilize fine-grained proppants with low proppant concentration (generally 1.46–3.91 kg/m²) to enhance fracture conductivity [31]. The proppant is a 20–40 mesh ceramic proppant. The experimental scheme for changing the fracture height is as follows: the fracture heights are 23, 36, and 50 mm, with a proppant concentration of 4 kg/m². The rock slabs provided by the laboratory can only be processed into sizes of 23, 36, and 50 mm. The length-to-height ratio of fractures was designed based on field observations and theoretical frameworks widely adopted in unconventional reservoir engineering. Specifically, we focused on a range of 0.14 to 0.35 for this ratio, which aligns with typical fracture geometries observed in shale and tight sandstone reservoirs under similar stress conditions. The proppant is a 20–40 mesh ceramic proppant. The experimental scheme for changing the proppant diameter is as follows: the fracture height is 50 mm; the proppant concentration is 4 kg/m²; and the proppants are 20–40, 40–70, and 80–100 mesh ceramic proppants. For the experimental scheme involving different types of proppants, only the proppant type is substituted with quartz sand (original: ceramic proppant). The experimental scheme for other fracture structures is similar to that of the "−" type fracture.

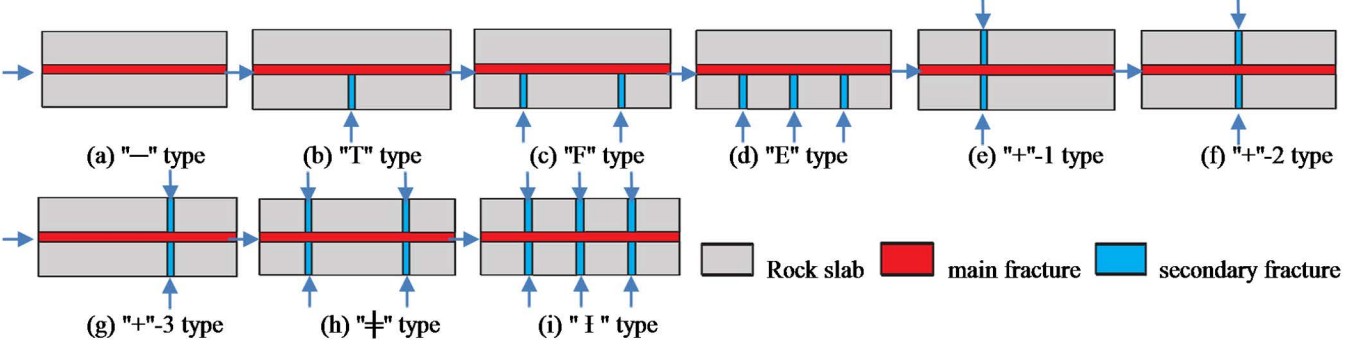

**Fig 3. Schematics of different types of fracture networks.**

Experimental procedure: Cut the rock slab and combine it into a fracture network. Place a specific amount of proppant in the fractures. Seal the gap and unpropped fracture with sealant, and install the sealing plate in the measurement chamber. Connect the inlet and outlet pressure ports, along with the flow measurement ports, to the chamber using standardized fluidic connectors. Pressurize the diversion chamber with closure pressures of 10, 20, 30, 40, 50, and 60 MPa.

## 3 Results and discussion

### 3.1 Experimental results of "—type" fracture network

Fig 4 shows the conductivity of the fracture network under different closure pressures and proppant concentrations. The trend of the fracture network's conductivity with closure pressure is similar across different proppant concentrations. As closure pressure increases, the conductivity of the fracture network decreases. When the proppant concentration is 4 kg/m², the conductivity is 12.446 μm²·cm at 10 MPa, decreases to 10.681 μm²·cm at 30 MPa, and drops to 2.228 μm²·cm at 60 MPa. Based on the analysis of six sets of curves, the decrease in conductivity under high closure pressure (60 MPa) is 81.7% − 85.1% compared to low closure pressures (10 MPa). At low closure pressure, the proppant stays intact. The large pores formed by the proppant result in high permeability and greater fracture conductivity. The decrease in conductivity is due to the elastic deformation of the proppant, which slightly reduces pore size. At high closure pressure, the proppant breaks. The broken ceramic proppant has low porosity, resulting in lower permeability and significantly reduced fracture conductivity. This experimental investigation employed high-brittleness rock slabs exhibiting elevated Young's modulus characteristics. There is less proppant embedment. The fracture conductivity is greater than that of rock slabs with low Young's modulus. The closure pressure is generally the minimum principal stress. The smaller the minimum stress is, the greater the fracture conductivity will be.

From Fig 5, it can be observed that an increase in closure pressure leads to a decrease in fracture conductivity. The conductivity of the fracture network exhibits exponential growth with increasing height. To increase fracture height in the deep part of the fracture, it's necessary to use high displacement and high viscosity fracturing fluid to carry the proppant, forming a high and long propped fracture to improve the conductivity.

Fig 6 shows that the conductivity of the fracture network increases with the proppant diameter. The 20–40 mesh proppant has the highest conductivity, while the 80–100 mesh proppant has the lowest due to its smaller particle diameter. The difference in fracture conductivity among the three proppant diameters is significant under low-pressure conditions but less pronounced under high-pressure conditions. Due to the complex structure and various fracture widths of the fracture

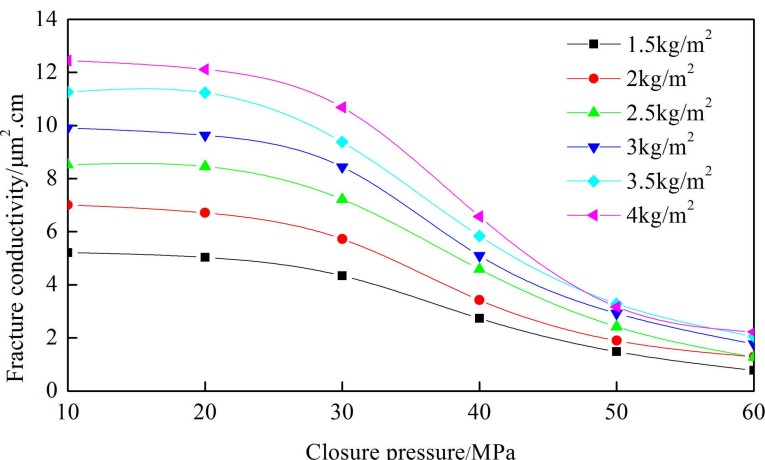

**Fig 4. Fracture conductivity of "−" fracture network with different ceramic concentrations.**

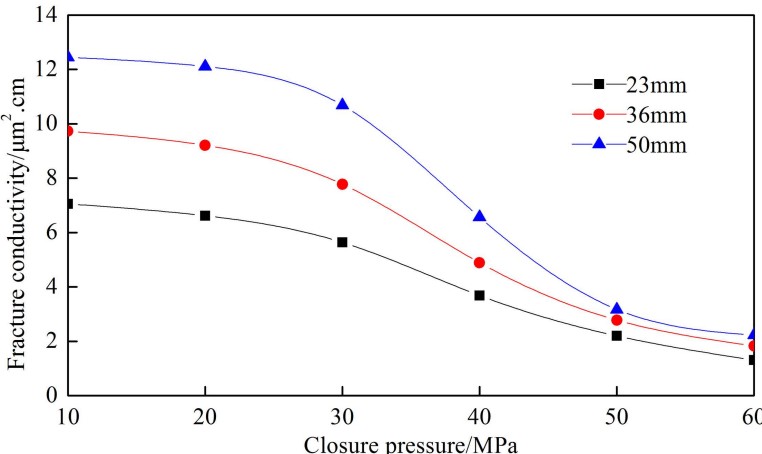

**Fig 5. Fracture conductivity of "−" fracture network with different fracture heights.**

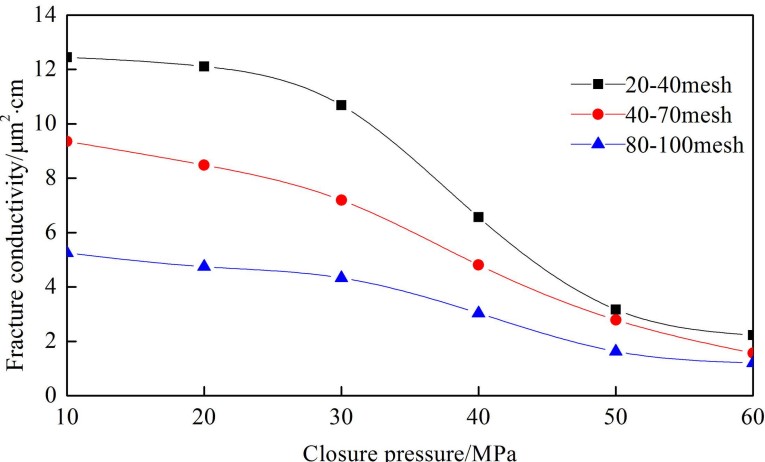

**Fig 6. Fracture conductivity of "−" fracture network with different proppant diameters.**

network, larger proppant cannot easily enter the small fractures. Based on the experimental results, the optimal proppant diameter for volume fracturing is 40–70 mesh, which has a smaller decrease in conductivity compared to the 20–40 mesh proppant and can be carried into the narrow fractures.

Comparing Figs 4 and 7, the fracture conductivity of quartz sand at 10 MPa is slightly higher than that of ceramic proppant. However, under high pressure, it is lower than that of ceramic proppant. The fracture conductivity decreases by 92.1% with the closure pressure increasing, indicating that the quartz sand possesses lower conductivity. It can be concluded that when the closure pressure is greater than 30 MPa, quartz sand cannot effectively provide higher conductivity and is unsuitable for high closure pressure conditions.

### 3.2 The influence of fracture network structure on conductivity

Figs 8–12 show the conductivity of different fracture networks under different conditions.
The conductivity of different types of fracture networks is analyzed under the same conditions.

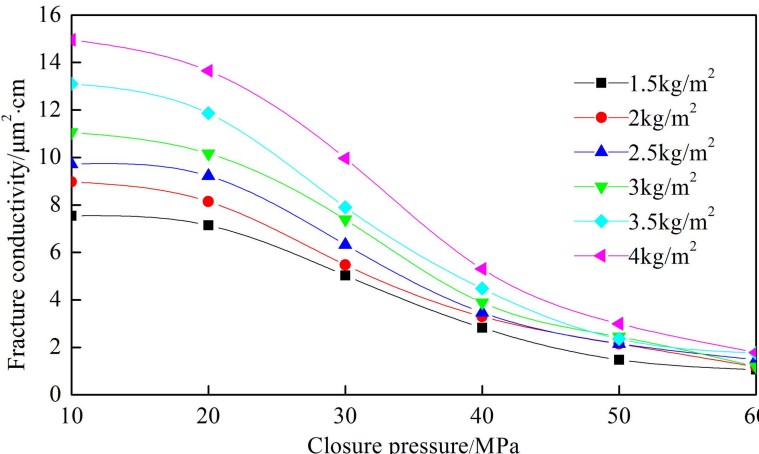

**Fig 7. Fracture conductivity of "−" fracture network with different quartz proppant concentrations.**

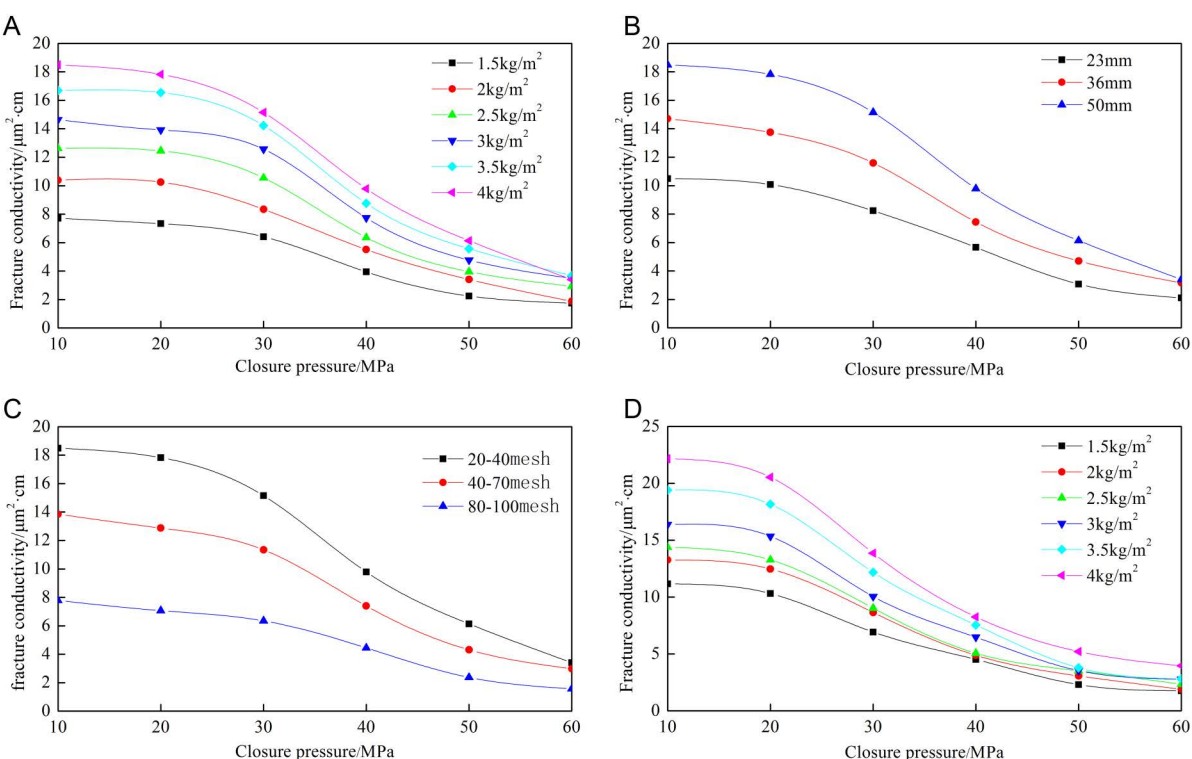

**Fig 8. Experimental results of "T" fracture network.** (a) Concentration of ceramic proppant, (b) Fracture height, (c) Proppant diameter, (d) Concentration of quartz sand.

Table 1 demonstrates that compared to the "− type" fracture network, the initial fracture conductivity continuously increases with the number of secondary fractures. The growth factors of the fracture network conductivity are 1.49, 2.31, 2.50, 2.88, and 3.01 for the "T", "F", "E", "╪", and "╫" types, respectively. The ratio of the final conductivity (60MPa) to the

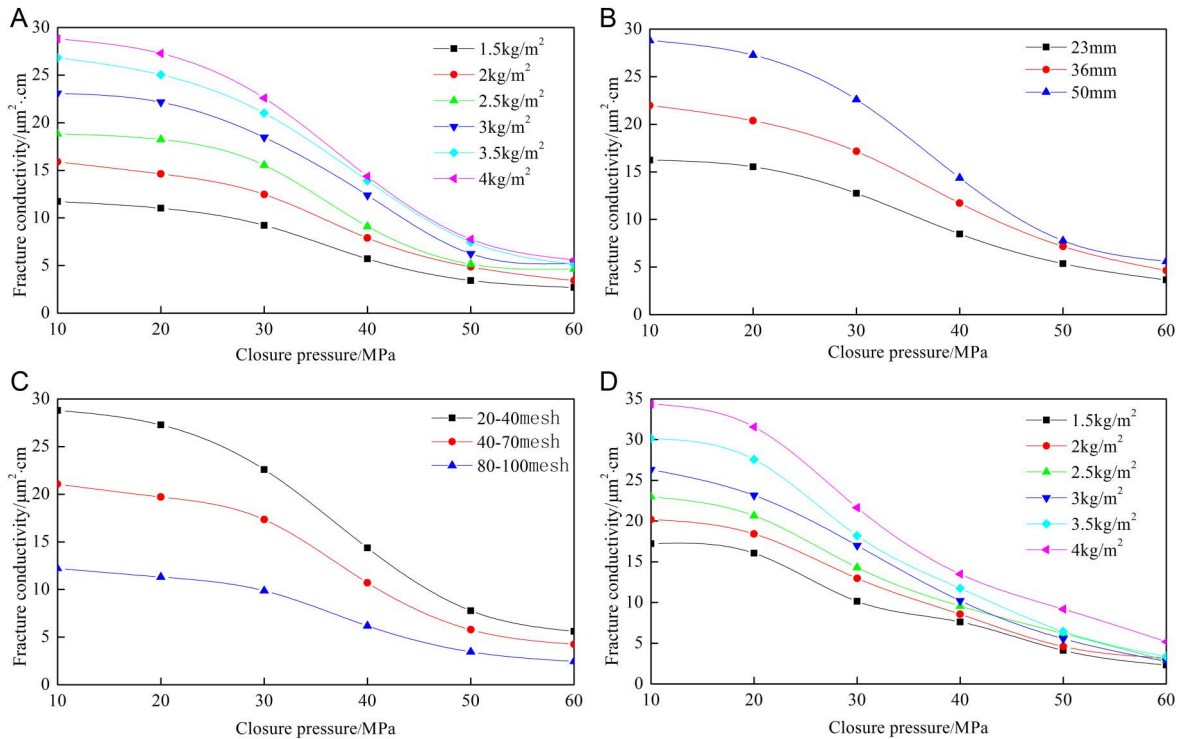

**Fig 9. Experimental results of "F" fracture network.** (a) Concentration of ceramic proppant, (b) Fracture height, (c) Proppant diameter, (d) Concentration of quartz sand.

initial conductivity (10MPa) also indicates that more secondary fractures lead to greater final fracture conductivity. Sufficient secondary fractures under high closure pressure can ensure that the formation maintains a certain degree of fracture conductivity.

Table 2 reveals that the nearer the secondary fractures are to the outlet of the chamber, the more significant the increase in the conductivity of the fracture network. From the ratio of the final conductivity (60MPa) to the initial conductivity (10MPa), it can be observed that the closer the secondary fractures are to the outlet, the greater the conductivity of the fracture network under high closure pressure. The closer the secondary fracture is to the outlet, the smaller the resistance during the flow process and the lower the energy loss in the secondary fractures. When the fluid enters the main fracture, the closer it is to the outlet, the lower the energy of the fluid in the main fracture, which means that the energy loss of mixing the fluid in the main and secondary fractures is the smallest and the increase in fracture conductivity is the largest.

Using the 20−40 mesh ceramic proppant, with a proppant concentration of 4 kg/m² and a fracture height of 50 mm, fracture conductivity obtained from nine types of fracture networks was plotted in Fig 13. The fracture conductivity exhibits the following descending order: "—" type < "+"-1 type < "T" type < "+"-2 type < "F" type < "E" type < "+"-3 type < "╪" type < "╫" type. The following observations have been found: (1) The conductivity of the "—" and "+"-1 fracture networks is close to each other and exhibit the lowest conductivity. Although the "+"-1 fracture network has two secondary fractures, their locations are far away from the outlet, resulting in high pressure loss during flow and significant energy loss when the fluid converges from the secondary fractures into the main fracture. (2) The conductivity of the "+"-2 fracture network is slightly higher than that of the "T" fracture network. The secondary fracture positions are the same, but the "+"-2 fracture network has an additional secondary fracture. (3) The conductivity of the "F" and "E" fracture networks is similar and significantly improved. Both have one secondary fracture closest to the outlet, indicating that the position of the secondary fracture

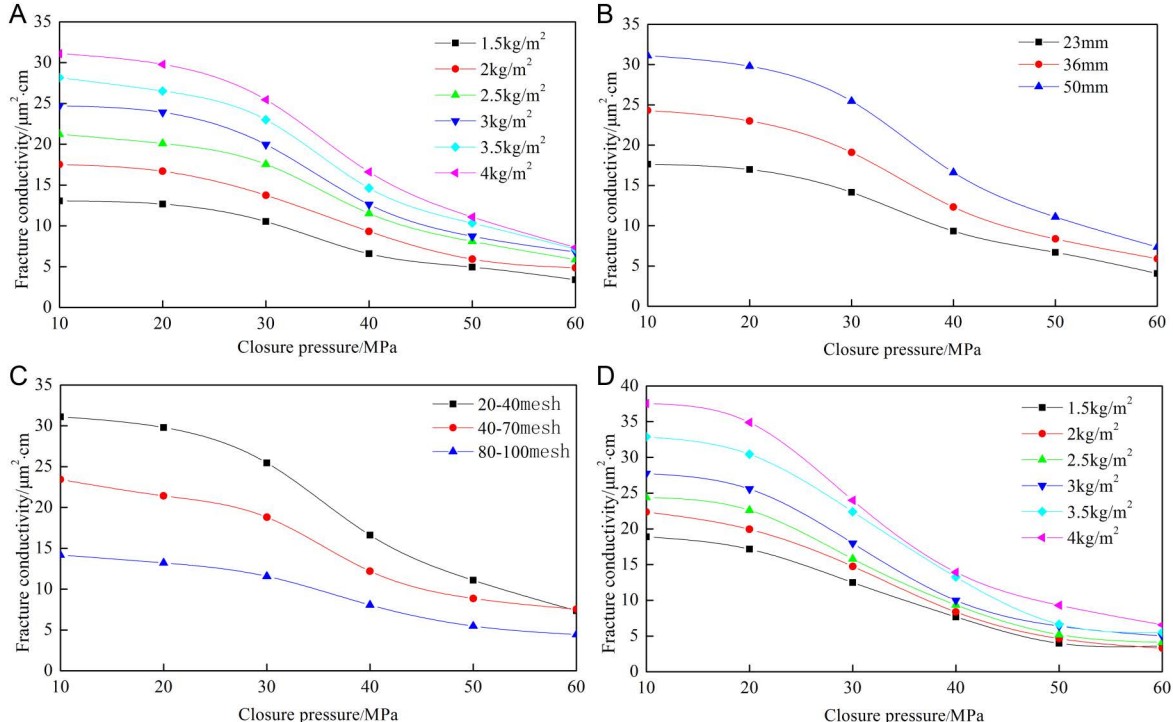

**Fig 10. Experimental results of "E" fracture network.** (a) Concentration of ceramic proppant, (b) Fracture height, (c) Proppant diameter, (d) Concentration of quartz sand.

significantly impacts the conductivity. (4) The conductivity of the "+"-3, "╪", and "╫" types of fracture networks is the highest because they have two secondary fractures closest to the outlet. The "╫" type has 6 secondary fractures; the "╪" type has 4; and the "+"-3 type has 2. Therefore, the "╫" type has the highest fracture conductivity, followed by the "╪" type, and the "+"-3 type has the lowest among the top three.

### 3.3 Orthogonal experimental analysis

This study considers seven influencing factors with the assumption of no interactions between them. Among these factors, the proppant type is assigned two distinct levels, while the remaining six factors each have three levels. Through orthogonal array reference tables, the optimal experimental layout is identified as $L_{18}(2 \times 3^7)$. The designed orthogonal experiments are implemented to quantify fracture conductivity variations, with detailed results provided in Table 3.

The F critical value is selected according to the F value critical value table with a confidence interval of 0.01. According to the verification results in Table 4, the closure pressure and the position of secondary fractures have the greatest impact on the conductivity of the fracture network. The proppant diameter and the proppant concentration also significantly affect fracture conductivity, while the number of secondary fractures, proppant type, and fracture height exhibit a weaker impact.

Under high closure pressure, the proppant is crushed, leading to a significant reduction in fracture width and pore size, and sharply decreasing permeability and fracture conductivity. The structure of the fracture network, including the number and position of secondary fractures, significantly impacts fracture conductivity, especially the position of secondary fractures. Fracture conductivity increases as secondary fractures approach the wellbore vicinity, with closer positioning minimizing fluid mixing-induced energy losses under equivalent fracture. Under comparable spatial configurations, an increase in secondary fracture number directly enhances fracture network conductivity. The flow diversion effect created by secondary

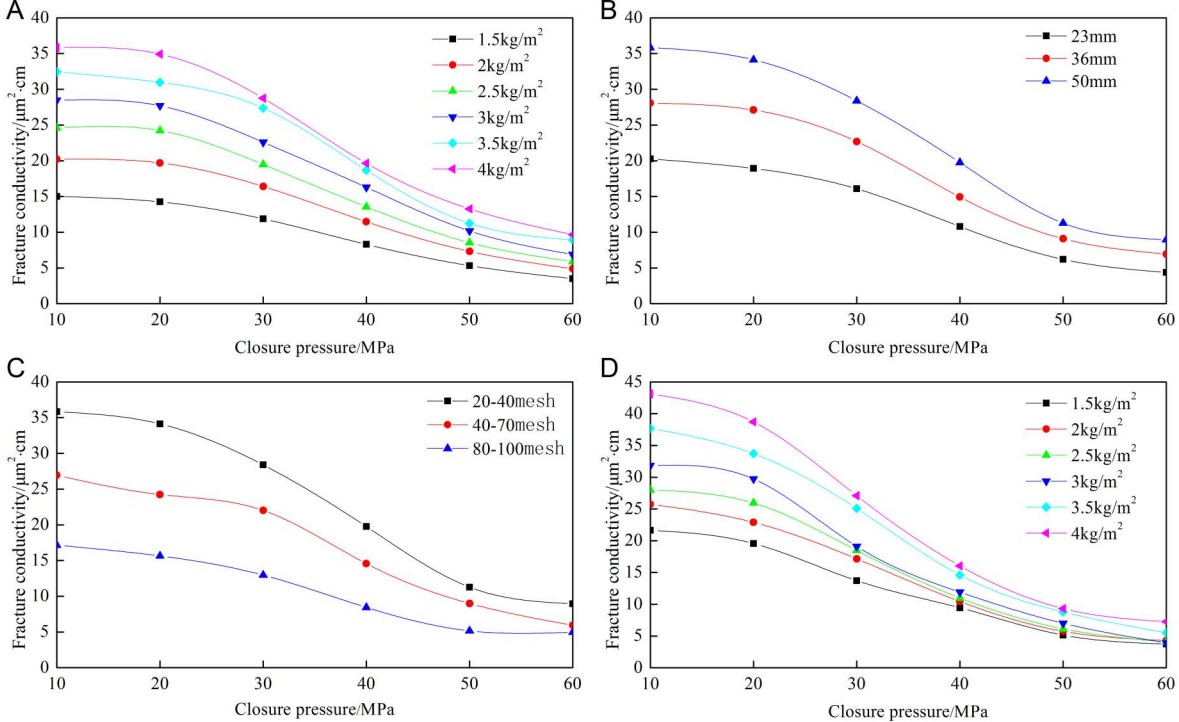

**Fig 11. Experimental results of "╈" fracture network.** (a) Concentration of ceramic proppant, (b) Fracture height, (c) Proppant diameter, (d) Concentration of quartz sand.

fractures significantly mitigates gas flow energy dissipation within the fracture network, thereby enhancing fracture conductivity. A larger proppant concentration results in a wider fracture width, which significantly enhances fracture conductivity. Under the same conditions, higher fracture height increases fracture conductivity, but the impact is not as significant as the discussed five conditions. The influence of the type of proppant on fracture conductivity is the weakest.

## 4 Analytical investigation of fracture network conductivity

Based on experimental data of fracture network conductivity and the discretized fracture network model, the sand layer inside the fracture is divided into two types: the sand layer on the fracture wall and the sand layer inside the fracture. Considering factors such as proppant embedment, fragmentation, and deformation, an analytical model of single fracture conductivity is first established. Leveraging the hydraulic-electrical analogy principle, an analytical model for fracture network conductivity is rigorously derived through multi-physical coupling analysis. The accuracy analysis of the analytical model is carried out.

Certain simplified assumptions are necessary regarding the internal gas flow in the fracture network and the complex sand laying method of the proppant:

1. The shape of the fracture network is fishbone-like;

2. Gas flow is Darcy flow;

3. Energy loss caused by gas collisions is neglected;

4. Secondary fractures are orthogonal to the main fracture;

5. The fluid flow in the rock matrix is neglected.

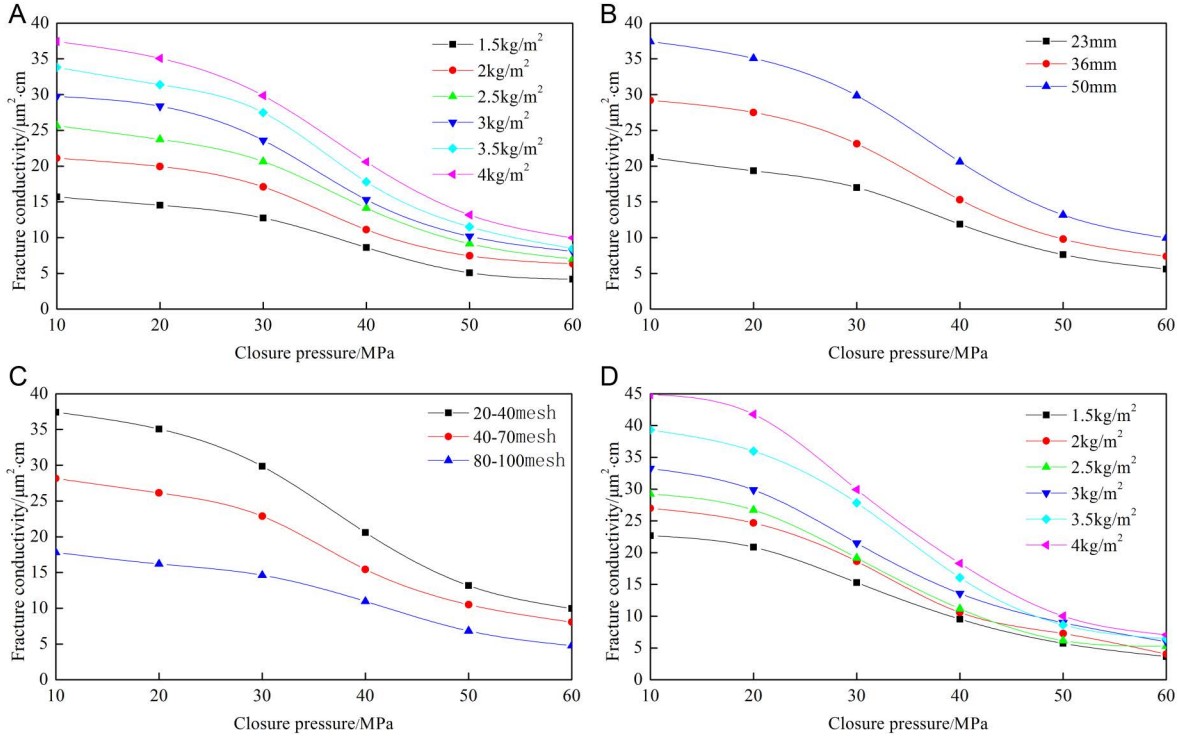

**Fig 12. Experimental results of "卝" fracture network.** (a) Concentration of ceramic proppant, (b) Fracture height, (c) Proppant diameter, (d) Concentration of quartz sand.

**Table 1. Influence of secondary fracture number on fracture conductivity.**

| Fracture network | "T" type | "F" type | "E" type | "╪" type | "卝" type |
|---|---|---|---|---|---|
| Number of secondary fractures | 1 | 2 | 3 | 4 | 6 |
| Initial fracture conductivity growth factor | 1.49 | 2.31 | 2.50 | 2.88 | 3.01 |
| Final conductivity/Initial conductivity | 18.4% | 19.5% | 23.7% | 26.9% | 29.3% |

**Table 2. Influence of secondary fracture position on fracture conductivity.**

| Fracture network | "+"-1 type | "+"-2 type | "+"-3 type |
|---|---|---|---|
| Distance from air outlet/mm | 120 | 80 | 40 |
| Initial fracture conductivity growth factor | 1.20 | 1.66 | 2.82 |
| Final conductivity/Initial conductivity | 20.6% | 22.1% | 24.9% |

## 4.1 A new model of the conductivity of fracture network

The basic assumptions of the analytical model of the single fracture conductivity are as follows: The proppant is a rigid sphere that does not deform or break; the proppant is arranged in a prismatic pattern. Based on the Kozeny-Carman model [32,33], the proppant sand layer is treated as a capillary model (Fig 14). The new model assumes that all capillaries are horizontal and uniform. In reality, both horizontal and vertical capillaries exist within the proppant. At low pressure, gas predominantly mainly flows in horizontal directions. Under high closure pressures, proppant crushing increases gas flow through vertical capillaries, leading to a decrease in the model's accuracy.

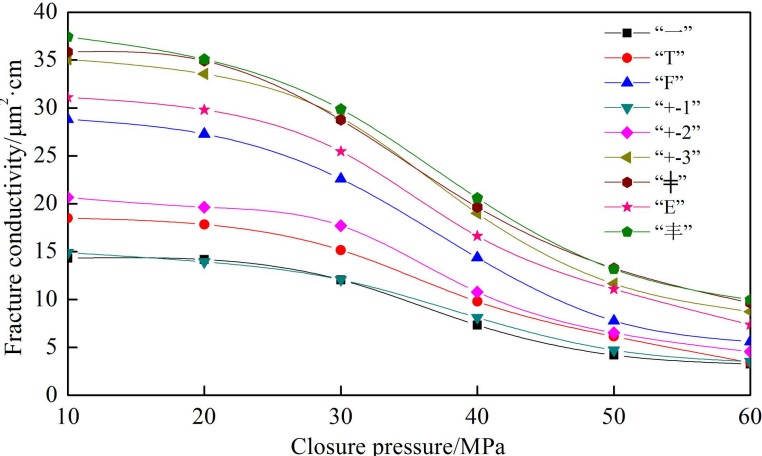

**Fig 13. Comparison of the conductivity of nine types of fracture network.**

**Table 3. Results of the orthogonal experiments.**

| Number | Proppant type | Number of secondary fractures | Distance from air outlet/mm | Fracture height/mm | Proppant concentration/(kg/m²) | Proppant diameter/mesh | Closure pressure/MPa | Error | Fracture Conductivity /μm²·cm |
|---|---|---|---|---|---|---|---|---|---|
| 1 | Ceramic proppant | 1 | 120 | 23 | 2 | 20-40 | 20 | 1 | 4.623 |
| 2 | | 1 | 80 | 36 | 3 | 40-70 | 40 | 2 | 5.027 |
| 3 | | 1 | 40 | 50 | 4 | 80-100 | 60 | 3 | 2.994 |
| 4 | | 2 | 120 | 23 | 3 | 40-70 | 60 | 3 | 1.215 |
| 5 | | 2 | 80 | 36 | 4 | 80-100 | 20 | 1 | 6.948 |
| 6 | | 2 | 40 | 50 | 2 | 20-40 | 40 | 2 | 10.936 |
| 7 | | 4 | 120 | 36 | 2 | 80-100 | 40 | 3 | 2.073 |
| 8 | | 4 | 80 | 50 | 3 | 20-40 | 60 | 1 | 6.875 |
| 9 | | 4 | 40 | 23 | 4 | 40-70 | 20 | 2 | 15.804 |
| 10 | Quartz sand | 1 | 120 | 50 | 4 | 40-70 | 40 | 1 | 6.305 |
| 11 | | 1 | 80 | 23 | 2 | 80-100 | 60 | 2 | 1.147 |
| 12 | | 1 | 40 | 36 | 3 | 20-40 | 20 | 3 | 17.339 |
| 13 | | 2 | 120 | 36 | 4 | 20-40 | 60 | 2 | 1.674 |
| 14 | | 2 | 80 | 50 | 2 | 40-70 | 20 | 3 | 6.624 |
| 15 | | 2 | 40 | 23 | 3 | 80-100 | 40 | 1 | 3.402 |
| 16 | | 4 | 120 | 50 | 3 | 80-100 | 20 | 2 | 6.378 |
| 17 | | 4 | 80 | 23 | 4 | 20-40 | 40 | 3 | 8.627 |
| 18 | | 4 | 40 | 36 | 2 | 40-70 | 60 | 1 | 2.774 |

The pores formed in the fracture can be divided into two types (Fig 15): one type is the pores near the fracture wall, which can be partially affected by the proppant embedding; the other type is the internal pores formed by the proppant, which is influenced by the proppant diameter.

(1) Calculation of the number of sand layers:

$$h = c/\rho_p \qquad (3)$$

**Table 4. Analyses of the variances.**

| Source of variance | Sum of squares | Free degree | Mean sum of squares | F value | F critical value | Significance |
|---|---|---|---|---|---|---|
| Proppant type | 0.275 | 1 | 0.275 | 0.088 | 3.46 | |
| Number of secondary fractures | 11.536 | 2 | 5.768 | 1.851 | 3.11 | |
| Distance from air outlet | 80.685 | 2 | 40.343 | 12.945 | 3.11 | High saliency |
| Fracture height | 2.631 | 2 | 1.315 | 0.422 | 3.11 | |
| Proppant concentration | 19.490 | 2 | 9.745 | 3.127 | 3.11 | Saliency |
| Proppant diameter | 61.517 | 2 | 30.758 | 9.870 | 3.11 | Saliency |
| Closure pressure | 140.412 | 2 | 70.206 | 22.527 | 3.11 | High saliency |
| Error | 9.349 | 3 | 3.116 | | | |

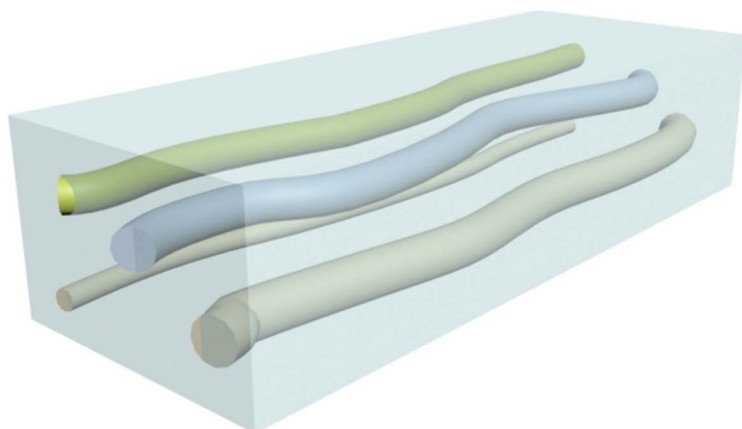

**Fig 14. Schematic of the capillary porous rock physical model.**

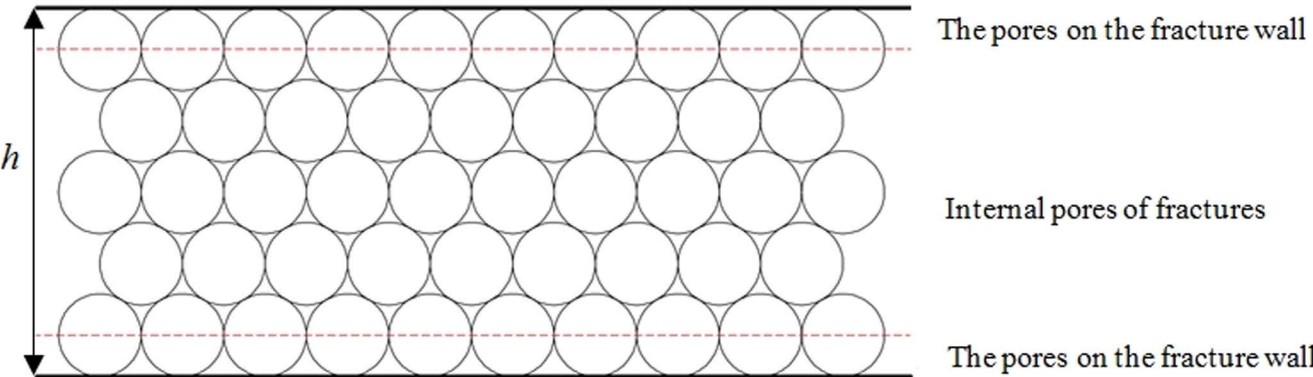

**Fig 15. Schematic diagram of proppant bedding pores.**

Where $c$ is the proppant concentration (kg/m²), $\rho_p$ is the bulk density of the proppant (kg/m³), $h$ is the proppant stacking height (m).

$$n = \left( \frac{c/\rho_p - 2r_p}{\sqrt{3}r_p} + 1 \right)$$

(4)

$$m = (n)_{int}$$

(5)

Where $n$ is a number of sand layers, $r_p$ is the proppant radius (m), $m$ is the integer of $n$.

(2)    Calculation of the proppant layers inside the fracture and the proppant layer near the fracture wall

Use $N$ to represent the total number of proppants in the sand layer. The placement method of proppants is cumulative layer by layer, with the same number placed in each layer:

$$N = \left[ n \left( \frac{H - 2r_p}{\sqrt{3}r_p} + 1 \right) - A \right] \left( \frac{L}{2r_p} \right)$$

(6)

Where $H$ is the fracture height (m), $L$ is the fracture length (m), $A$ is constant, $N$ is total number of proppant placement (integer). $m = 3i - 2, A = 2i - 2, m = 3i - 1, A = 2i - 1, m = 3i, A = 2i; m、I、A=1、2、3……$
    Due to the difference between the proppant layers adjacent to the fracture wall and other proppant layers, its layer number needs to be calculated.

$$N_f = \left[ 2 \left( \frac{H - 2r_p}{\sqrt{3}r_p} + 1 \right) - B \right] \left( \frac{L}{2r_p} \right)$$

(7)

Where $N_f$ is the number of proppants adjacent to the fracture wall (integer), $B$ is constant. $m = 3i - 2, B = 0, m = 3i - 1, B = 1, m = 3i, B = 1$.

(3) Calculation of fracture width

    The stacked proppant exhibits a triangular distribution under high closure pressure. The height of adjacent layers of proppant can be calculated by $2r_p\sqrt{6}/3$. Based on the fracture height, the fracture width can be evaluated:

$$w = (n - 1) 2r_p\sqrt{6}/3 + 2r_p$$

(8)

Where $w$ is the fracture width (m).

(4) Porosity calculation

    The porosity of the sand layer is the ratio of the pore volume of the sand layer to the total volume of the fractures supported by the proppant. According to the definition of porosity, the calculation equation for porosity is:

$$\phi = \frac{LHw - 4N\pi r_p^3/3}{LHw}$$

(9)

Where $\phi$ is the porosity of the main and secondary fracture.

## (5) Permeability calculation

Based on the assumption that the sand layer is viewed as a capillary model, the relationship between permeability and pore size is obtained using the Kozeny-Carman equation:

$$K = \frac{10^{12}\phi r^2}{8\tau^2} \tag{10}$$

Where $r$ is the pore radius supporting fractures (m), $\tau$ is tortuosity (based on this configuration, it is a constant of 1.1547), $K$ is the fracture permeability (μm²).

$$\tau = \left(\frac{L_w}{L}\right)^2 \tag{11}$$

Where $L_w$ is the actual length of the capillary tube (m).

Substitute the porosity calculation Equation 9 into 10 to obtain the single fracture permeability calculation equation, as follows:

$$K = \frac{\left[LHw - N\left(4\pi r_p^3/3\right)\right] r^2 \times 10^{12}}{8LHw\tau^2} \tag{12}$$

Calculation for pore radius of the propped fractures:

$$r = \frac{2}{n+1}r_1 + \frac{n-1}{n+1}r_2 \tag{13}$$

Where $r_1$ is the fracture wall pore radius (m), $r_2$ is the internal pore radius of fractures (m)

The pore channel near the fracture wall is enclosed by proppant hemisphere and the fracture plane, which is different from the pore channel inside the fracture. The calculation equation for the pore radius adjacent to the fracture wall is described as:

$$r_1 = \sqrt{\frac{2LHr_p - N_f\frac{2\pi r_p^3}{3}}{N_1\pi L}} \tag{14}$$

Where $N_1$ is the number of pores on the fracture wall surface.

The pores inside the fracture are composed of proppants, and their calculation equation is as follows:

$$r_2 = \sqrt{\frac{LH\left[(n-1)\,2r_p\frac{\sqrt{6}}{3}\right] - \left(N\frac{4\pi r_p^3}{3} - N_f\frac{2\pi r_p^3}{3}\right)}{N_2\pi L}} \tag{15}$$

Where $N_2$ is the number of internal pores in fracture.

$$N_1 = 2\left[(H-2r_p)/\left(\sqrt{3}r_p\right)\right] - C \tag{16}$$

$m = 3i-2, C = 0; m = 3i-1, m = 3i, C = 1.$

$$N_2 = (n-1)\left[(H-2r_p)/\left(\sqrt{3}r_p\right)\right] - D \tag{17}$$

$m = 3i - 2/m = 3i - 1, D = 2i - 2; m = 3i, D = 2i - 1.$

(6) Calculation of fracture conductivity

The equation for calculating the conductivity of a single fracture without considering embedding, proppant fragmentation, etc. is as follows:

$$Kw = \frac{\left[LHw - N\left(4\pi r_p^3/3\right)\right] r^2 \times 10^{12}}{8LH\tau^2} \tag{18}$$

The breakage of the proppant is not considered in the analytical model of the conductivity of a single fracture. Based on experimental data from the "− type" fracture network, the analytical model for single fracture conductivity is modified.

The z-values are the ratio of experimentally measured conductivity to the calculated ones; the ratio of closure pressure to proppant compressive strength is treated as x; the ratio of proppant radius to 20–40 mesh proppant radius is plotted as the y-axis (Fig 16).

$$f = 0.1946e^{-\left(\frac{P_c/\sigma_c - 0.227}{0.56}\right)^2}\left(1.932\frac{r_{20/40}}{r_p} - 0.9304\right) \tag{19}$$

Where $f$ is the correction factor, $P_c$ is the closure pressure (MPa), $\sigma_c$ is the proppant compressive strength (MPa), $r_{20/40}$ is the average radius of 20–40 mesh proppant (m), $r_p$ is the average radius of proppant (m).

The modified equation for calculating the conductivity of a single fracture is as follows:

$$(Kw)_f = \frac{\left[LHw - N\left(4\pi r_p^3/3\right)\right] r^2 \times 10^{12}}{8LH\tau^2} \cdot 0.1946e^{-\left(\frac{P_c/\sigma_p - 0.227}{0.56}\right)^2}\left(1.932\frac{r_{20/40}}{r_p} - 0.9304\right) \tag{20}$$

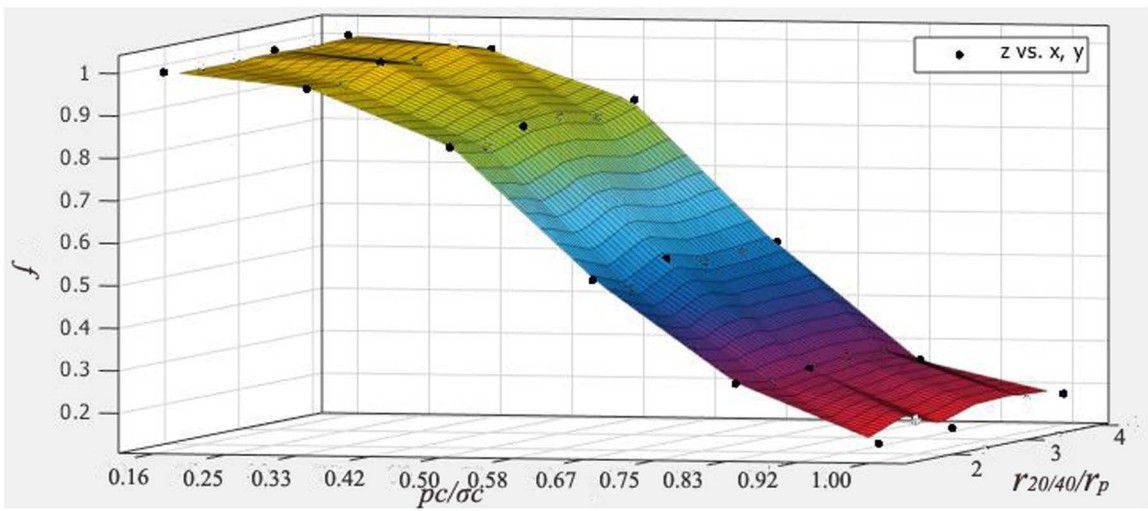

**Fig 16. Results of the correction factor.**

Where $(Kw)_f$ is the conductivity of a single fracture considering the closure pressure and proppant fragmentation ($\mu m^2 \cdot cm$).

The principle of the hydraulic-electrical analogy is that the seepage law of fluid flow in porous media has a certain similarity to the law of charge flow in conductive materials. These two physical processes share similar forms of governing equations and flow states.

During the experiment, a fracture network is constructed using rock slabs and proppants, as shown in Fig 17.

Due to the equal inlet pressures of the main and secondary fractures, the incoming gas enters the fracture network from the main fracture $R_1$ and secondary fractures $R_2$, $R_3$, $R_5$, $R_6$, $R_8$, and $R_9$. The flow rate inside the fracture is inversely related to the seepage resistance. The gas flowing into the main fracture $R_1$ and secondary fractures $R_2$ and $R_3$ converge at the intersection point and then flows into the main fracture $R_4$. Gas flows in other fractures, $R_5 \sim R_{10}$, follow similar patterns that are similar to the series-parallel current flow. In a parallel circuit, the potential is equal, and the current magnitude is inversely related to the resistance. In a series circuit, the current is equal, and the potential is additive (Fig 18).

The total flow rate inside the chamber remains unchanged, and the inlet and outlet flow rates are equal. This is similar to the equal current in series-parallel circuits. The outlet pressure is affected by the structure of the fracture network and seepage resistance, similar to how the negative electrode potential in a circuit is affected by resistance and the structure of the circuit. Seepage resistance is influenced by factors like permeability, fracture length, and fracture width, while electrical resistance is influenced by resistivity, resistance length, and cross-sectional area. The two are similar in this regard.

The analytical model of fracture network conductivity is established based on a single fracture conductivity model. Firstly, using the modified single fracture conductivity model (Equation (3–20)), the conductivity of the main and secondary fractures is obtained. Then, using Equation (21), the seepage resistance $R_1 \sim R_{3j+1}$ of the main and secondary fractures is calculated.

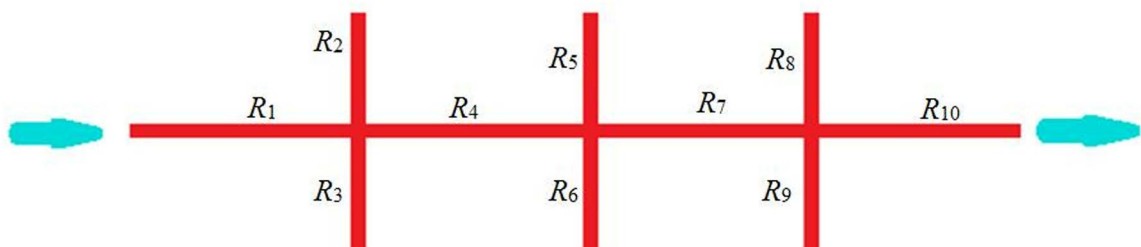

**Fig 17. Schematics of "†" fracture network.**

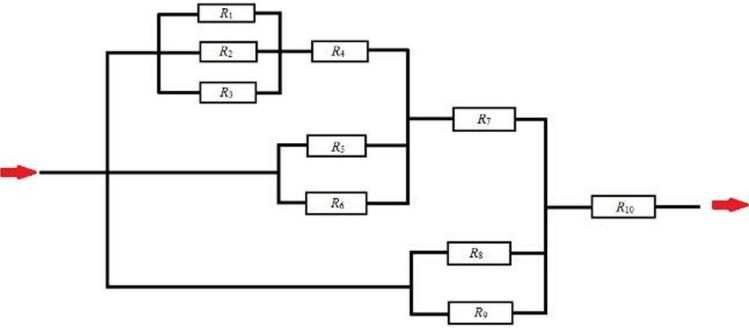

**Fig 18. Equivalent circuit diagram of "†" fracture network.**

$$R_k = \frac{\mu L}{(Kw)_f H} \ (k = 1, 2, 3 \ldots \ldots 3j+1)$$

(21)

Where $\mu$ is the viscosity of air under standard condition, 0.01789mPa.s, $R_k$ is the seepage resistance (mPa·s/ (μm².cm)).

According to the equivalent circuit diagram and Ohm's law, the seepage resistance is calculated. When the secondary fracture is closing or the proppant concentration is zero, the seepage resistance of the crack is infinite.

As shown in Fig 19, there are 2$j$ secondary fractures, and the general equation for calculating the seepage resistance is shown::

$$R_{t1} = \frac{1}{\frac{1}{R_1} + \frac{1}{R_2} + \frac{1}{R_3}} (j = 1)$$
$$R_{tj} = \frac{1}{\frac{1}{R_{t(j-1)} + R_{3j-2}} + \frac{1}{R_{3j-1}} + \frac{1}{R_{3j}}} (j \geq 2)$$
$$R = R_{tj} + R_{3j+1}$$

(22)

According to Equation 22 obtain the seepage resistance of the complex fracture network, and then use Equation 23 to obtain the permeability of the complex fracture network.

$$K_{nw} = \frac{\mu L}{RwH}$$

(23)

Where $K_{nw}$ is the permeability of the complex fracture network (μm²).

$$F_{nwc} = K_{nw}w_f$$

(24)

Where $F_{nwc}$ is the conductivity of the complex fracture network (μm².cm).

### 4.2 Verification of the fracture network conductivity model

Fig 20 demonstrates strong congruence between the analytical model-predicted fracture network conductivity and experimental measurements. The error for different proppant concentrations is about 13%. When the closure pressure is less than 50 MPa, the root mean square error is 5.82%. Under high closing pressure, the error is relatively large, but the difference is small, and the error is within a reasonable range.

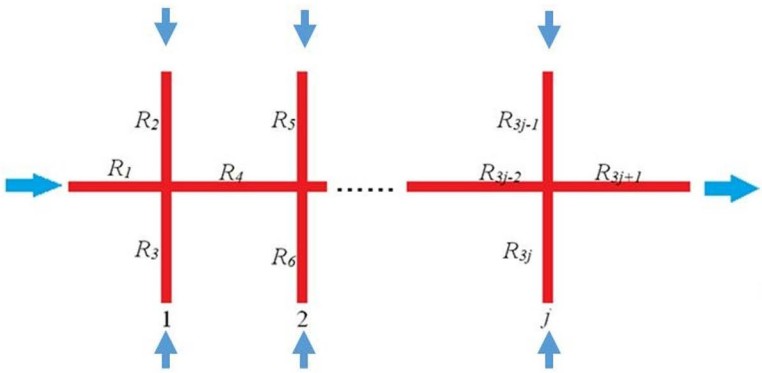

**Fig 19. Schematics of shark's fin shape fracture network.**

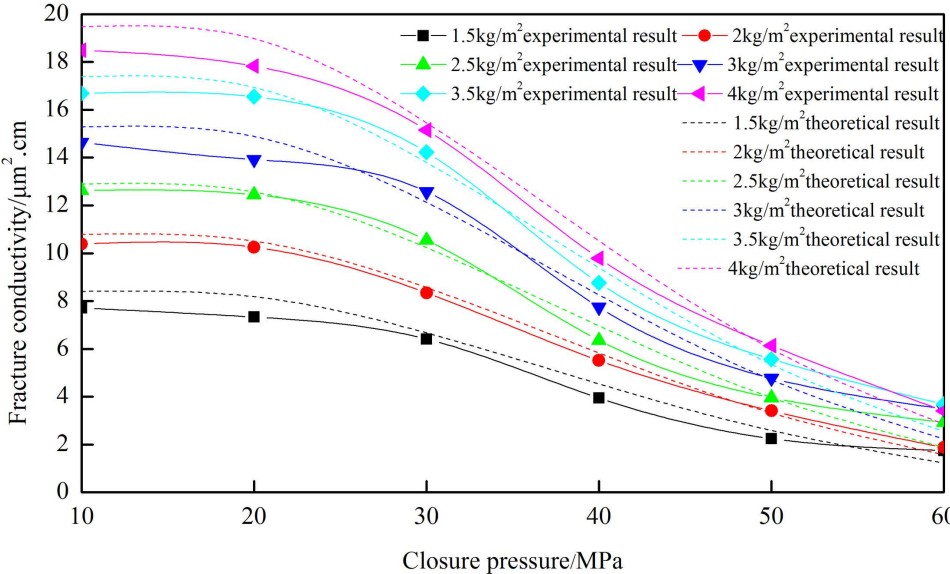

**Fig 20. Comparison of theoretical and experimental results for different concentrations of "T" type.**

Fig 21 reveals that the theoretical calculation results still remain in good agreement with the experimental data curves under different fracture heights. The error between theoretical and experimental results across different fracture heights is 13.16%, indicating high accuracy. When the closure pressure is below 50 MPa, the theoretical and experimental results exhibit excellent consistency with minimal deviation, with an error of 5.53%.

From Fig 22, the root mean square error for different proppant diameters is 11.27%, which meets the engineering accuracy requirement. Under low closure pressure conditions, the root mean square error is 3.96%, demonstrating high precision.

Fig 23 shows that under identical conditions but different fracture network structures, the theoretical model results exhibit strong agreement with the experimental data. The root mean square error is 9.81%, with the primary source of error attributed to lower measurement accuracy under high closure pressure (60 MPa). At low closure pressures, the root mean square error decreases to 8.20%.

The new model established in this study demonstrates high accuracy, with a root mean square error below 14%, meeting the requirement for engineering applications. The theoretical results show excellent agreement with experimental data when the closure pressure is below 50 MPa. However, the accuracy decreases under high closure pressures, primarily due to limitations in the precision and reliability of experimental measurements under such conditions. Additionally, the new model assumes that all capillary flow pathways are horizontally oriented, whereas, in reality, both horizontal and vertical capillaries exist within the proppant. At low pressure, gas predominantly flows along horizontal directions with minimal vertical movement. Under high closure pressures, proppant crushing increases gas flow through vertical capillaries, causing the model to underestimate experimental results and resulting in larger deviations.

## 4.3 Application of the fracture network conductivity model

The fracture conductivity of different numbers of secondary fractures is calculated based on the new model.

As illustrated in Fig 24, the conductivity of the fracture network exhibits a positive correlation with secondary fracture density. The increase of conductivity caused by the increase in the number of secondary fractures is 5%,

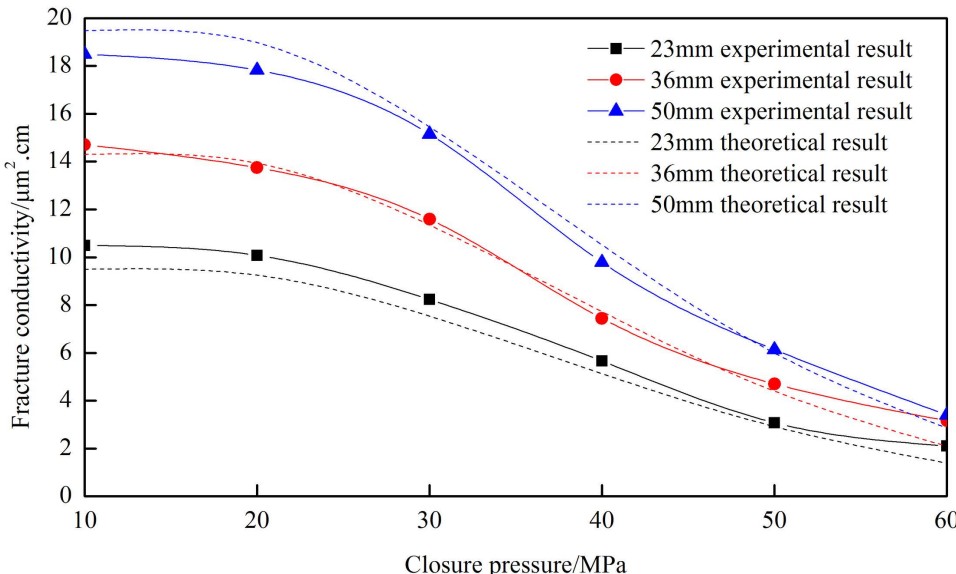

**Fig 21. Comparison of theoretical and experimental results for different fracture heights of "T" type.**

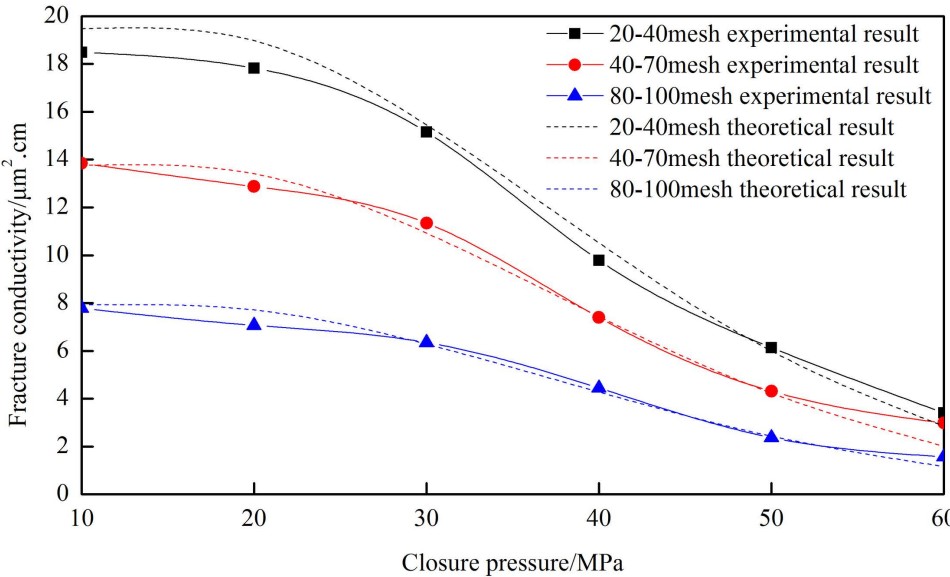

**Fig 22. Comparison of theoretical and experimental results for different proppant diameters of "T" type.**

29.63%, 8.79%, 49.89%, and 15.45%, respectively. Secondary fracture exhibits a positive correlation with matrix permeability, as increased fracture networks provide augmented seepage conduits and correspondingly diminished flow resistance.

The aspect ratio is defined as the ratio of the main fracture length to the secondary fracture length. This parameter governs the geometry of the fracture network and serves as a critical parameter in hydraulic fracturing design. In this study, aspect ratios of 2, 3, 4, 5, and 6 are investigated.

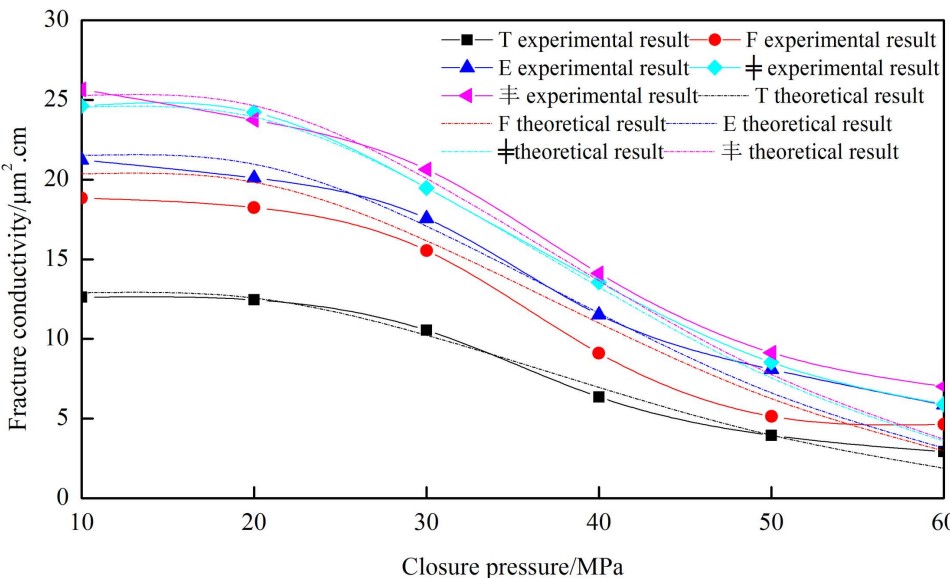

**Fig 23. Comparison of theoretical and experimental results under different fracture structures.**

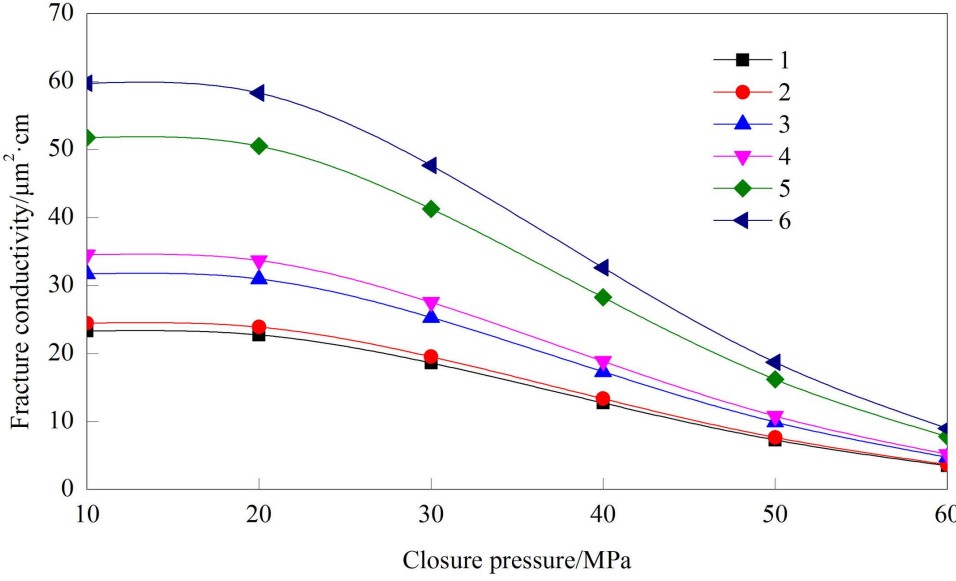

**Fig 24. The conductivity of fracture network under different numbers of secondary fractures.**

Fig 25 reveals that the conductivity of the fracture network increases with the aspect ratio. However, the rate of increase diminishes as the aspect ratio rises. At a closure pressure of 20 MPa, when the aspect ratio increases from 2 to 3, the conductivity rises from 39.82 µm²·cm to 45.10 µm²·cm, representing a 13.26% improvement. In contrast, when the aspect ratio increases from 5 to 6, the conductivity only improves from 52.04 µm²·cm to 54.49 µm²·cm, corresponding to a smaller gain of 4.61%. These results indicate that while higher aspect ratios enhance conductivity, the incremental gain

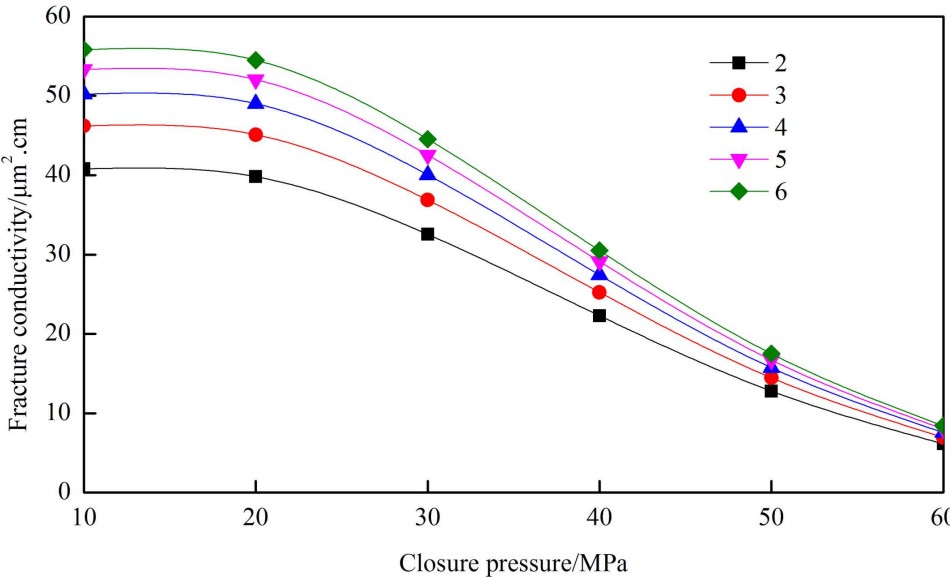

**Fig 25.  Conductivity of fracture network under different aspect ratio.**

becomes progressively smaller. For optimal fracture network design in practical applications, an aspect ratio ranging from 4 to 5 is generally recommended.

## Conclusions

In this study, experiments are conducted to investigate the effect of critical influential factors on the conductivity of the fracture network. A new model of fracture network conductivity is established. The following conclusion can be drawn:

1. The experiments show that as closure pressure increases, the conductivity of the fracture network decreases. Quartz sand proppant shows a similar trend, but the fracture conductivity is lower compared with ceramic proppant.

2. The higher the proppant concentration, the greater the conductivity. As the fracture height and proppant diameter increase, the conductivity increases. The closer the position of the secondary fracture is to the outlet, the greater the increase in the conductivity of the fracture network.

3. The fracture conductivity exhibits the following descending order: "-" type＜"+"-1 type＜"T" type＜"+"-2 type＜"F" type＜"E" type＜"+"-3 type＜"╪" type＜"╫" type. Orthogonal experimental analysis shows that closure pressure and the position of secondary fractures significantly impact the conductivity of the fracture network. The proppant diameter and proppant concentration also have a greater impact, while the number of secondary fractures, type of proppant, and fracture height have a weaker impact.

4. An analytical model of single fracture conductivity is established based on the Kozeny-Carman equation and experimental data. Based on the principle of hydraulic-electrical analogy, a new model of the fracture network conductivity is ultimately established.

5. For fracture network fracturing optimization, it is recommended to use smaller proppant particle sizes (40–70 mesh) with lower proppant concentration and high-strength proppant. In fracture network design, an aspect ratio of 4–5 is preferred, and multiple secondary fractures should be designed near the wellbore to enhance fracture conductivity.

## Supporting information

**S1 Table.  Original data.**

(XLSX)

## Author contributions

**Data curation:** Jinjian Gao.

**Formal analysis:** Jinjian Gao.

**Writing – original draft:** Jinjian Gao.

**Writing – review & editing:** Lanxiao Hu.

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
