## [Decision Letter · Decision Letter 0]

14 Mar 2025

Dear Dr. gao,

Thank you for submitting your manuscript to PLOS ONE. After careful consideration, we feel that it has merit but does not fully meet PLOS ONE’s publication criteria as it currently stands. Therefore, we invite you to submit a revised version of the manuscript that addresses the points raised during the review process.

Please address the comments from two reviewers and improve the quality of the paper. 

We look forward to receiving your revised manuscript.

Kind regards,

Jianguo Wang, PhD

Academic Editor

PLOS ONE

Journal Requirements:

“the Natural Science Foundation of Sichuan Province (2023NSFSC0945)”

Reviewers' comments:

Reviewer's Responses to Questions

**Comments to the Author**

1. Is the manuscript technically sound, and do the data support the conclusions?

Reviewer #1: Yes

Reviewer #2: Yes

2. Has the statistical analysis been performed appropriately and rigorously?

Reviewer #1: Yes

Reviewer #2: Yes

3. Have the authors made all data underlying the findings in their manuscript fully available?

Reviewer #1: Yes

Reviewer #2: No

4. Is the manuscript presented in an intelligible fashion and written in standard English?

Reviewer #1: Yes

Reviewer #2: Yes

Reviewer #1: The effects of fracture network structure, fracture height, proppant concentration, proppant particle size, and proppant type on the conductivity of the fracture network are studied by experiments and orthogonal analysis. I think this article deserves to be published in Research Article. The current analysis method is simple and easy to use. I think this research is very valuable. Here are some suggestions for improving this research:

1. The control range of some experimental variables can be further expanded when studying the influence of different factors on the conductivity of fracture networks. For example, more proppant types or wider closure pressure intervals can be added to obtain more comprehensive data.

2. In the process of model validation, the comparison analysis with actual gas reservoir production data can be added in addition to the comparison with experimental data.

3. This paper does not fully discuss the influence of complex geological conditions (such as different rock types, stress distribution, etc.) on the conductivity of fracture networks.

4. The experimental scheme does not fully explain the selection basis of sand concentration, fracture height, proppant particle size, and other factors.

5. In the part of model validation, the adaptability of the model under different fracture network structures and different working conditions is not fully analyzed.

Reviewer #2: The paper demonstrates innovation in experimental design and model construction, with sufficient data volume. However, methodological details, model validation, and language expression require refinement. Revisions can further enhance scientific rigor and application value.

Reviewer Comments:

1、The paper designs a novel flow diversion chamber and establishes a fracture network conductivity model, showing some innovation. However, the abstract should more explicitly emphasize its breakthrough points compared to existing studies.

2、The selection of flow chamber material (stainless steel) lacks justification for whether its compressive performance meets the requirement of 100 MPa closure pressure. Please supplement material mechanical parameters.

3、The orthogonal experiment does not clarify the rationale for selecting specific factor levels (e.g., fracture heights of 23/36/50 mm). Provide basis for parameter selection.

4、The Kozeny-Carman equation assumes the proppant pack as a uniform capillary tube model, but actual fracture networks may have non-uniform distributions. Discuss potential impacts of this assumption on model accuracy.

5、Figure 20 only validates the "T-shaped" fracture network. Include comparisons with other types (e.g., "I-shaped," "E-shaped") to demonstrate universality.

6、The model exhibits significant errors (13%) under high closure pressure, yet the error sources are not thoroughly analyzed. Add discussion on error origins.

7、The critical F-value criteria in Table 4 are undefined. Specify the statistical testing basis.

8、The conclusion lacks clarity on how the findings guide field fracturing optimization (e.g., optimal proppant size selection, fracture network design). Include engineering application recommendations.

9、Author abbreviations in some references (e.g., Refs. 23, 27) are inconsistent. Standardize formatting.

10、Some sentences contain grammatical errors (e.g., "Due the complex structure" should be "Due to..."). Perform full-text proofreading.

**Do you want your identity to be public for this peer review?** For information about this choice, including consent withdrawal, please see our Privacy Policy

Reviewer #1: No

Reviewer #2: No

---

## [Author Response · Author response to Decision Letter 1]

23 Mar 2025

First and foremost, I would like to express my sincere gratitude for the thorough review and insightful comments provided on our manuscript titled " Experimental and theoretical investigation on the conductivity of complex fracture network in unconventional gas reservoirs ". Those comments are all valuable and very helpful for revising and improving our paper, as well as the important guiding significance to our researches. We have carefully considered each point raised and have made the following revisions to our manuscript.

---

## [Decision Letter · Decision Letter 1]

7 May 2025

Dear Dr. gao,

Thank you for submitting your manuscript to PLOS ONE. After careful consideration, we feel that it has merit but does not fully meet PLOS ONE’s publication criteria as it currently stands. Therefore, we invite you to submit a revised version of the manuscript that addresses the points raised during the review process.

**ACADEMIC EDITOR:**

This manuscript comes to the last step. Please fully check both English and presentations.A partially marked version is attached for your reference. 

We look forward to receiving your revised manuscript.

Kind regards,

Jianguo Wang, PhD

Academic Editor

PLOS ONE

Journal Requirements:

Reviewers' comments:

Reviewer's Responses to Questions

**Comments to the Author**

Reviewer #1: All comments have been addressed

2. Is the manuscript technically sound, and do the data support the conclusions?

Reviewer #1: Yes

3. Has the statistical analysis been performed appropriately and rigorously?

Reviewer #1: Yes

4. Have the authors made all data underlying the findings in their manuscript fully available?

Reviewer #1: Yes

5. Is the manuscript presented in an intelligible fashion and written in standard English?

Reviewer #1: Yes

Reviewer #1: The authors correctlt answered my comments, and I recommend that this paperr can be accepted as it is.

**Do you want your identity to be public for this peer review?** For information about this choice, including consent withdrawal, please see our Privacy Policy

Reviewer #1: No

---

## [Author Response · Author response to Decision Letter 2]

15 May 2025

Dear Editor,

Thank you for guiding us through the final stage of manuscript preparation. We sincerely appreciate your emphasis on rigorous language and presentation standards.

Academic Editor

Comment: This manuscript comes to the last step. Please fully check both English and presentations. A partially marked version is attached for your reference..

Response: The authors thank the academic editor for carefully reading. Dr. Hu, the second author, who has four years of academic experience in Canada, was invited to check and revise the English language. The manuscript has undergone comprehensive formatting adjustments aligning with the journal's style specifications. We have double-checked the manuscript and corrected the grammar errors in the revised manuscript. We also have revised our manuscript thoroughly, and the language is refined to make the manuscript more concise and logical. The linguistic expressions have been refined into more academically rigorous English formulations

Changes: Please see the revised manuscript.

We would be honoured to implement any additional specific adjustments per your expert guidance. Thank you for your invaluable support in advancing this work.

Sincerely

Jinjin Gao

Corresponding Author

---

## [Editor Report · Decision Letter 2]

20 May 2025

Experimental and theoretical investigation on the conductivity of complex fracture network in unconventional gas reservoirs

PONE-D-25-07188R2

Dear Dr. gao,

We’re pleased to inform you that your manuscript has been judged scientifically suitable for publication and will be formally accepted for publication once it meets all outstanding technical requirements.

Kind regards,

Jianguo Wang, PhD

Academic Editor

PLOS ONE
---

## [Editor Report · Acceptance letter]

PONE-D-25-07188R2

PLOS ONE

Dear Dr. gao,

I'm pleased to inform you that your manuscript has been deemed suitable for publication in PLOS ONE. Congratulations! Your manuscript is now being handed over to our production team.

Kind regards,

on behalf of

Dr. Jianguo Wang

Academic Editor

PLOS ONE